# The Novel Digital Therapeutics Sensor and Algorithm for Pressure Ulcer Care Based on Tissue Impedance

**DOI:** 10.3390/s23073620

**Published:** 2023-03-30

**Authors:** Tae-Mi Jung, Dae-Jin Jang, Jong-Ha Lee

**Affiliations:** 1Department of Biomedical Engineering, School of Engineering, Keimyung University, Daegu 42601, Republic of Korea; 2Industry-Academic Cooperation Foundation, Keimyung University, Daegu 42601, Republic of Korea

**Keywords:** pressure ulcer care device, biophotonic sensor

## Abstract

Visual diagnosis and rejuvenation are methods currently used to diagnose and treat pressure ulcers, respectively. However, the treatment process is difficult. We developed a biophotonic sensor to diagnose pressure ulcers and, subsequently, developed a pressure ulcer care device (PUCD.) We conducted animal and clinical trials to investigate the device’s effectiveness. We confirmed the accuracy of the pressure ulcer diagnosis algorithm to be 91% and we observed an 85% reduction in immune cells when using the PUCD to treat pressure ulcer-induced mice. Additionally, we compared the treatment group to the pressure ulcer induction group to assess the PUCD’s effectiveness in identifying immune cells through its nuclear shape. These results indicate a positive effect and suggest the use of PUCD as a recovery method for pressure ulcer diagnosis and treatment.

## 1. Introduction

International classification systems established by the European Pressure Ulcer Advisory Panel and the National Pressure Injury Advisory Panel are widely used for classifying pressure ulcers in four stages [1]. The European Pressure Ulcer Advisory Panel states that the prevalence and incidence of pressure ulcers among individuals in aged care facilities worldwide are 32.2% and 59.0%, respectively [2]. Bedsores lower the patient’s quality of life, and as a result, they complain of physical and mental pain until they are cured [3]. In addition, they feel the burden of medical expenses according to the treatment and management of pressure ulcers. Therefore, the early detection and treatment of pressure ulcers is important.

Currently, to diagnose pressure ulcers, doctors observe the affected area with the naked eye. This has a varying effect, depending on the skill level of the doctor. According to previous studies, pressure ulcer classification system education programs were based on hospital nurses’ knowledge of incontinence associated dermatitis, as well as visual discriminability [4]. Since nurse expertise around pressure ulcers was not well developed, diagnoses regarding the condition were not accurate, leading to serious errors in the prevention and treatment of pressure ulcers [2]. Diagnosis by medical staff with the naked eye is not accurate and there is a need for an alternative method to improve the accuracy of pressure ulcer diagnoses.

A second method for the early diagnosis of pressure ulcers is to use skin pressure and temperature sensors [5]. The pressure sensor continuously measures the pressure applied to areas prone to pressure ulcers depending on measured body conditions. If pressure is applied to a specific body part for a prolonged period, the smart device worn by the caregivers receives warnings, leading to prompt corrective action.

A third method of diagnosis is with a system that predicts the different stages of pressure ulcers, uses AI with pictures, and recommends appropriate dressing agents [6]. The device uses fluorescent imaging, a functional imaging method, to obtain images at several consecutive wavelengths with a fluorescent lamp as a light source. The image is generated using the fluorescent characteristics of intrinsic and extrinsic molecules. Accordingly, normal and tumor regions are typically differentiated via their pixel brightness. Malignant cells absorb sensitizers at a higher rate, resulting in increased fluorescence when compared with the normal surrounding skin. Further, red fluorescence, emitted by the activated precursor, is visualized and diagnosed with a Wood’s lamp.

A fourth method of diagnosis is with biopsy and histological observation of the skin. However, infection risk and inconvenience in sample management after examination are some of the drawbacks.

With skin imaging and diagnosis, the equipment used for point-of-care measurement suffers from image distortion, because image acquisition is performed through surface contact with the skin. Moreover, conventional imaging systems do not consider differences in relative skin tones and color values between patients during analysis, causing errors. Since these devices comprise single-mode hardware, the number of diagnosable skin lesions is also limited. In addition, their results are not reproducible due to contact measurement.

The treatment of pressure ulcers requires the periodic changing of positions to prevent prolonged pressure on the skin, and repeated washing and dressing of the affected area. This treatment method is expensive as it needs periodic management by healthcare workers and caregivers.

Laser therapy, a treatment method for pressure ulcers, is used for effective wound treatment and local pain recovery. Low-intensity laser treatment is typically used to treat wounds, including ulcers such as pressure ulcers, with many theoretical hypotheses and benefits reported. Low-intensity lasers are most effective for patients with pressure ulcers [7,8,9,10] as they activate blood circulation and promote the migration of inflammatory cells and wound regeneration. Additionally, they are relatively low-powered compared to surgical lasers.

Ultraviolet light treatment, another method of phototherapy, has been reported to significantly reduce the buttock lesions caused by stage two pressure ulcers when treating patients with stage two to four pressure ulcers [1]. The therapeutic effects of ultraviolet rays prevent the progression of lesions in early-stage pressure ulcers by increasing the blood supply to the skin, promoting cell growth, and sterilizing the skin. The type and intensity of laser or ultraviolet light can be manipulated according to the state of the pressure ulcer lesions during pressure ulcer intervention. These treatment methods have been shown to promote wound healing, pain relief, and the sterilization of pressure ulcers.

Open wet dressing therapy uses food wraps to clean the wound and maintain moisture without disinfection for pressure ulcer treatment over a short period of time [11]. It is an efficient treatment method compared to conventional methods. However, the periodic changing of dressings is a significant downside. Moreover, the lack of experience among the medical staff in applying the dressing is a significant concern. The method requires extensive practice to ensure familiarity, and cutting the dressing to fit the wound appropriately is not easy. Therefore, people other than healthcare workers would be unable to change the dressing.

Additionally, new therapeutic agents, extracorporeal shock wave therapy and infrared therapy, have been reported to be effective treatment methods.

Several researchers have reported the use of extracorporeal shock wave therapy to treat soft tissue damage and reduce or completely heal skin wounds. The mechanism of treatment for extracorporeal shock wave therapy has not been identified.

In addition, a pressure ulcer treatment device that provides negative pressure therapy has been developed [12]. It applies pressure to the wound and suctions it to create new capillaries in the wound, and automatically removes pus.

The effects of the shockwave generation method, stimulation intensity and frequency, and treatment frequency on pressure ulcers have not been identified owing to a lack of studies.

Infrared treatment also has adverse effects such as dry treatment areas and the risk of exposure to glass fragments when the light bulb ruptures. Further, its efficacy is limited to a smaller area compared to conversion heat treatment.

Since negative pressure therapy devices are typically used for stage three and stage four pressure ulcers, they are not suitable for treating pressure ulcers where the treatment and diagnosis in the early stages are very important.

Therefore, diagnosing and treating pressure ulcers by addressing the drawbacks of these existing methods is necessary. This study aims to develop a quantitative diagnosis algorithm for pressure ulcers using the biophotonic sensor developed in previous studies and we propose the diagnosis and treatment of pressure ulcers using a pressure ulcer care device (PUCD) based on this technology.

## 2. System and Algorithm

### 2.1. Development of a Pressure Ulcer Care Device

In this study, a biophotonic sensor was used to develop a biophotonic PUCD for diagnosing and treating pressure ulcers using light and by measuring impedance change [11]. The prototype consists of a power supply, an operating unit and control, a display, and light irradiation units. A rechargeable internal battery powers the skin impedance electrode and biophotonic sensor.

The components of the PUCD are as follows: the pins responsible for impedance oscillation are each surrounded by six pins responsible for receiving the biopotential signal. Accordingly, a total of 18 impedance values are calculated from three cells. Figure 1a shows the arrangement of the pins, and Figure 1b shows the top view of the PUCD. The corrected signal value, obtained by calculating the change between the impedances of the oscillation and receiving pins for each cell, is stored on the server. Six red LEDs of a wavelength of 660 nm in a circle around each cell allow the treatment to proceed according to the diagnostic result based on the impedance signal value and position of the affected area. Then, 18 red lights are set to irradiate with a biophotonic sensor with an applied light intensity between 0 and 100%. Furthermore, two methods are applied to minimize skin irritation at the time of diagnosis. First, a gold (Au) coating is applied to the outside of the impedance pins that meet the skin during diagnosis so that the current can flow well for each affected part. Second, the inside of the impedance pin has a spring to reduce skin irritation.

The red lights in the biophotonic sensor are placed in a hexagon for optimal light intensity and consider the overlapping of light to ensure effective light distribution. The selected red-light wavelength of 660 nm ± 5% [5,10,13,14] is absorbed well into the tissues, promoting metabolic cell activity and activating cell regeneration to treat skin disease [15,16]. The appearance applied to the skin can be confirmed through Figure 1c.

### 2.2. Impedance Measurement Algorithm

The biophotonic sensor used in the PUCD can identify areas with high and low impedances via electromyogram and impedance analysis modules. This enables the determination of pressure ulcer severity by generating EMG and impedance maps based on parameters such as skin moisture content, density, and elasticity [17]. A pressure ulcer diagnosis algorithm, used for severity determination, uses algorithm-based data mining for classification, which is trained on 200 impedance measurements.

To evaluate the accuracy of the pressure ulcer diagnosis algorithm, the data preprocessing filter was changed from numeric to binary to convert all numeric attributes to binary before setting no class attributes. The accuracy of the pressure ulcer diagnosis algorithm was evaluated based on the accuracy of 10-fold cross-validation (the mean of the results of repeating the procedure to randomly select 80% of the 200 impedance data points as learning data and 20% as non-learning data 10 times).

The biophotonic sensor array was independently selected by the control hardware of each electrode to enable paired impedance measurements between the electrodes. Custom software implemented in Python communicated with the control hardware to select electrode pairs in a specific order and simultaneously control the LCR meter. The microcontroller routed the test signal from the LCR meter to the selected electrodes and recorded the impedance. Subsequently, the microcontroller selected the next set of electrodes in the measurement sequence. The impedance data were analyzed via custom MATLAB scripts using statistics, system identification, and control systems toolboxes. Minimizing contact impedance and stray capacitance is critical to obtaining accurate tissue impedance measurements. Therefore, the manufacturing process was optimized to minimize the contact impedance by selectively applying high-conductance gel to each electrode. Therefore, a patterned stencil and blade coating method was used. Appropriate pressure was applied during measurement to ensure sufficient contact of the measurement array with the tissue. Inadequate capacitance and other parasitic resistances were also minimized by choosing the minimum length of shielded cable to the LCR meter, shielded Bayonet Neill–Concelman connections, and noise-minimizing layout and routing techniques for the custom control hardware.

In this study, impedance data were generated by conducting preliminary and animal experiments to observe trends in the impedance values for pressure ulcers and normal regions using the PUCD. The test group was divided into three categories: normal, pressure ulcer, and non-conductive regions. The measurements were obtained on wet tissues and mouse pads simulating a human forearm and pressure ulcer area, respectively.

Further, the PUCD was used in animal experiments to set the optimal number of light parameters for pressure ulcer treatment. An evaluation was conducted using light optimization factors such as wavelength, time, and intensity. A preliminary study was conducted using four experimental wavelengths of 365, 455, 545, and 660 nm. The 660 nm near-infrared region wavelength was selected as it was not toxic to cells and induced cell proliferation. At this wavelength, the impedance frequency band was determined to be an important factor. To evaluate the impedance frequency band more accurately, 20–29, 50–59, and 90–99 kHz bands were tested to select the frequency band most similar to the impedance measurement value of the PUCD. A random 62 kΩ resistor was connected to channel seven of the device and its impedance was measured for the 20–29, 50–59, and 90–99 kHz frequency bands. Consequently, channel seven in the 20–29, 50–59, and 90–99 kHz bands was measured to be most similar at 61.7, 60.8, and 58.7 kΩ, respectively. Hence, the final frequency range of the PUCD was selected as 20–29 kHz. This measurement was conducted using a Wavelength Dispersive X-ray Fluorescence Spectrometer (XRF-1800, Shimadzu, Kyoto, Japan) at Intelligent Construction System Core-Support Center, Keimyung University, Republic of Korea.

### 2.3. Animal Experiment

Previous studies performed cell experiments to determine the effectiveness of phototherapy. As a result of confirming the wound healing ability by 660 nm light irradiation, it was confirmed that the light-irradiated group showed similar results to the VEGF used as a positive control group compared to the untreated group (Control) [18,19]. Through this result, it was confirmed that 660 nm light irradiation was effective in wound healing. Figure 2 and Figure 3 show the experimental results. In addition, in order to confirm the angiogenesis-promoting effect according to 660 nm light irradiation, it was confirmed that angiogenesis was induced faster in the light-treated group than in the untreated group through Tube Formation, which confirmed the angiogenic effect of HUVEC cells. Based on these results, we conducted animal experiments.

We conducted animal experiments with the approval of the Keimyung University Animal Experimental Performance Management Committee (KM-IACUC) with the approval number KM-2021-18R1. Twelve male laboratory rats (Sprague Dawley) aged 8–9 weeks old, weighing approximately 250–300 g, and raised under the same conditions, were used in the experiments. Isoflurane, an anesthetic, was administered at an inhalation concentration of 2–2.5% in oxygen with a vaporizer. After we removed the hair from the back, pressure ulcers were induced for two h at 120 mmHg using forceps. The animal experimental model used was presented in a thesis on the therapeutic efficacy of pressure ulcers via light irradiation [17]. The wavelength of the biophotonic sensor of the device used in the experiment was 660 nm, the time of application was 30 and 60 min, the intensity was 50% and 100%, and all 18 LEDs were operated.

The measurements were taken three times in five groups. Rats from groups one and two were reused in groups four and five. In group one, the rats were treated with the 660 nm light at 50% intensity for 30 and 60 min by dividing the pressure ulcer induction area in two. Figure 4 confirms that the areas causing pressure sores are divided. In group two, the rats were treated with 660 nm light, at 100% intensity, for 30 and 60 min by dividing the pressure ulcer induction area in two. In groups four and five, the pressure ulcer experiment was conducted by inflicting wounds with sandpaper on rats from groups one and two, respectively. In group three, the rats were treated with the 660 nm light at 100% intensity for 120 min (the highest possible dose). Before the induction of pressure ulcers, the impedance of the rats in each group was measured five times daily. After the pressure ulcers were induced, the impedance and light of each group were measured five times twice daily (morning and afternoon). Finally, pictures of the ulcer area were acquired during each measurement.

The rats were anesthetized using isoflurane mixed with oxygen (5% isoflurane for an oxygen inhalation concentration of 2–2.5%, nitrous oxide–oxygen ratio of 2:1). Anesthesia was maintained with 2% isoflurane for the oxygen inhalation concentration of 1.5–1.8% and a 2:1 ratio of nitrous oxide and oxygen. After the rat’s back was shaved and hair was removed using a hair removal cream over an area of 5 × 10 cm^2^, the skin was wiped with alcohol. Next, the skin was gently stretched, and forceps were used to apply 120 mmHg of pressure to the skin twice a day for two hours to induce pressure ulcers on the skin. In addition, sandpaper was additionally used to leave scratches on the skin, causing second-stage pressure sores. This can be seen in Figure 5.

After successfully inducing pressure ulcers, the biophotonic treatment was administered using the PUCD twice a day for 2 h. The rats were immobilized using anesthesia each time, and the treatment was administered for one to two hours daily over five to seven days. After treatment, euthanasia was induced using carbon dioxide (CO_2_) and pictures of the pressure ulcer sites were acquired. The method of applying PUCD by group is shown in Figure 6.

The rat skin tissues were collected and treated with 10% formalin to be analyzed. This process was sequentially performed on three animals in each group: normal, pressure ulcer-induced, untreated, and fluorescence-treated groups.

Next, tissue slides were prepared for histological analysis, and hematoxylin and eosin (H&E) and Masson’s trichrome staining were performed on the skin tissue to identify collagen fibers.

## 3. Results

### 3.1. Algorithm Accuracy Analysis

A total of 200 measurements were performed on normal and pressure ulcer areas with the PUCD. The overall range distribution for the normal and pressure ulcer areas was 600–1000 kΩ with an average of 714 kΩ, and 100–250 kΩ with an average of 173 kΩ, respectively. To analyze the impedance data, a graph was generated after data filtering and simple interpolation using MATLAB (version 8.2, MathWorks, Natick, MA, USA). This graph is shown in Figure 7 below. Based on the pressure ulcer data generated by this impedance mapping method, the intensity of the pressure ulcer was set to decrease as the impedance value of the affected area approached that of normal skin. The higher the impedance value of the affected area, the more the LED intensity needed to be adjusted. Based on the 2000 impedance values with a pressure ulcer area of 100× (impedance 1–10 channels) and normal area of 100× (impedance 1–10 channels), the accuracy analysis of the pressure ulcer diagnosis algorithm was 91%.

### 3.2. Animal Experiment

Through H&E staining, the degree of skin tissue damage caused by the first-step pressure sore induction and the degree of treatment effect by the pressure sore care device were both confirmed. The experimental results immediately before cell staining can be seen in Figure 8.

Compared to the normal group, epithelial loss, dermal tissue arrangement damage, fat layer irregularity, and muscle tissue damage were confirmed in the tissues induced by pressure sores. 

Following the application of PUCD onto the confirmed damaged tissue, the density of tissues, the shape of the fat layer, and the arrangement of muscles in the group treated with the pressure ulcer care device were closer to the normal group than the group without the PUCD. The degree of tissue damage can be confirmed through Figure 9, Figure 10 and Figure 11.

This confirmed the efficacy of the PUCD and the regeneration of the epithelial layer.

In addition, by counting the nuclei of the immune cells separately, the inflammatory response reduction effect on the stained tissue could be confirmed. These cells were identified by the shape of their nuclei by designating 15 identical regions on each tissue slide. About 85% of the immune cells were reduced in the PUCD treatment group compared to the inflammation induction group. A graph of the degree of immune cell reduction is shown in Figure 12.

## 4. Discussion

In this study, we developed a PUCD using a biophotonic sensor. To confirm its diagnostic accuracy and treatment effectiveness, an algorithmic analysis for accuracy was performed, confirming an accuracy of 91%. In the animal experiment, performed to verify the effectiveness of treatment, approximately 85% of immune cells were confirmed to be reduced via histological examination after treating the rats with the proposed PUCD. This implies that the treatment method facilitates pressure ulcer treatment to stages close to those of the normal group. The diagnostic accuracy was expected to be higher than the visual diagnosis, since it was performed using quantitative values.

A study of the phototherapeutic effect of low-level laser treatment on mitochondrial activity using a 660 nm laser, the same as our proposed PUCD, demonstrated a significant increase in mitochondrial activity (MTT) levels during both the inflammatory and regenerative phases [20,21].

Another study compared the effects of 660 nm and 808 nm low-level laser treatments on crush injuries in alveolar nerves and reported the 808 nm laser to be more effective [20]. This difference in the results was attributed to different species of mice being used for the experiments.

The specific threshold representing tissue damage in this study demonstrated the usefulness of impedance spectroscopy for the early detection of tissue damage. Since the measurement is made in contact with the skin, using impedance spectroscopy on stage one and stage two pressure ulcers is not expected to be an issue. However, using it periodically on stage three and stage four pressure ulcers is expected to make wound management difficult due to infection. We used the power of the second stage as the experimental condition, but additional research is needed to determine the optimal parameters by changing various power conditions (use more or less), parameter pulse, size, and frequency interval.

Pressure ulcers are diagnosed based on the subjective judgment of medical staff. With PUCD, it is possible to make a numerically accurate diagnosis regardless of the skill level of the healthcare worker. Furthermore, it is possible to treat pressure ulcers effectively by reducing the number of treatment steps through rapid diagnosis and accurate treatment according to progression. By reducing the treatment steps, the effect of reducing treatment costs can also be expected.

## 5. Conclusions

We proposed and developed a PUCD as an alternative to existing methods (such as visual examination, skin biopsy, and periodic dressing) that enables simple treatment and reduces costs for the diagnosis and treatment of pressure ulcers. We developed an algorithm to quantitatively diagnose pressure ulcers as well as a PUCD to apply the algorithm, using the biophotonic sensor developed in previous research. We analyzed the accuracy and followed this with an analysis of the benefits of the PUCD. We verified the direct effect of the PUCD on the pressure ulcer area during treatment. Additionally, we demonstrated an anti-inflammatory effect from the experiments using the PUCD. Epithelial tissue regeneration was also confirmed under an optical microscope. When the PUCD was used to treat pressure ulcers for 30 min, the tissue density, fat layer shape, and muscle arrangement were closer to those of the normal group than the control group.

PUCD is a multi-device that can immediately proceed with treatment according to the diagnosis result, and it is a faster and simpler process than existing diagnosis and treatment methods. It can improve the quality of the diagnosis and treatment of pressure ulcers by enabling the efficient management of pressure sore ulcers, which is cumbersome for patients, medical workers, and protectors.

The low-power laser treatment method using a biophotonic sensor is simple compared to conventional treatment methods and has fewer adverse effects as it does not significantly affect normal cells. Compared to the existing visual diagnosis method, our proposed device enables quantitative diagnosis and effective stepwise treatment, reducing the number of treatment steps and the costs. In addition, the device is compact, enabling its application in various areas. PUCD’s compactness facilitates the development of new healthcare equipment that exceeds the limitations of current pressure ulcer treatment.

In the future, if PUCD is mass-produced at low cost, it is expected that improved pressure ulcer treatment will become possible at home.

## Figures and Tables

**Figure 1 sensors-23-03620-f001:**
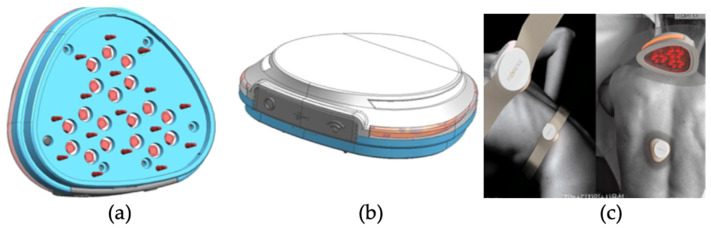
(**a**) Developed PUCD, (**b**) top view of developed PUCD, and (**c**) its application on the skin.

**Figure 2 sensors-23-03620-f002:**
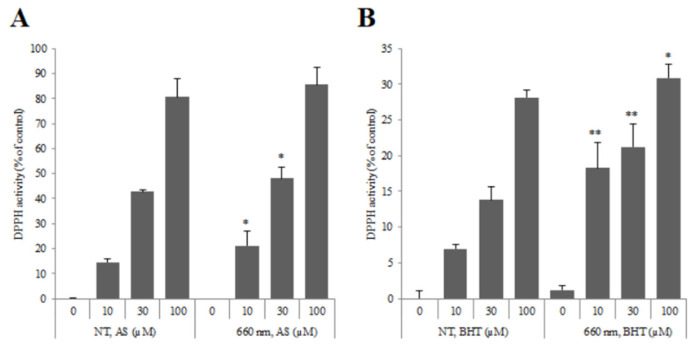
Verification of antioxidant efficacy according to 660 nm light irradiation. (**A**) is the result of pursuing demand activity in AS (Ascorbic acid) analysis, and (**B**) is the result of pursuing demand activity in BHT (Butylated hydroxytoluene) analysis. * means the significant difference when comparing the number of healthy cells in the population with the non-treted group at the *p* < 0.05 significance level, and ** at the *p* < 0.005 significance level.

**Figure 3 sensors-23-03620-f003:**
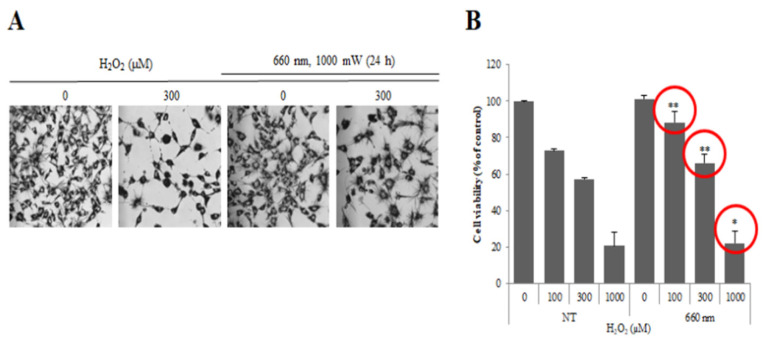
Confirmation of oxidative stress inhibition ability by 660 nm light irradiation. (**A**) is the result of confirming the efficacy of PUCD during apoptosis caused by oxidative stress. (**B**) is a graph comparing the cell viability of the PUCD-applied group and the non-treted group. * means the significant difference when comparing the number of healthy cells in the population with the non-treted group at the *p* < 0.05 significance level, and ** at the *p* < 0.005 significance level.

**Figure 4 sensors-23-03620-f004:**
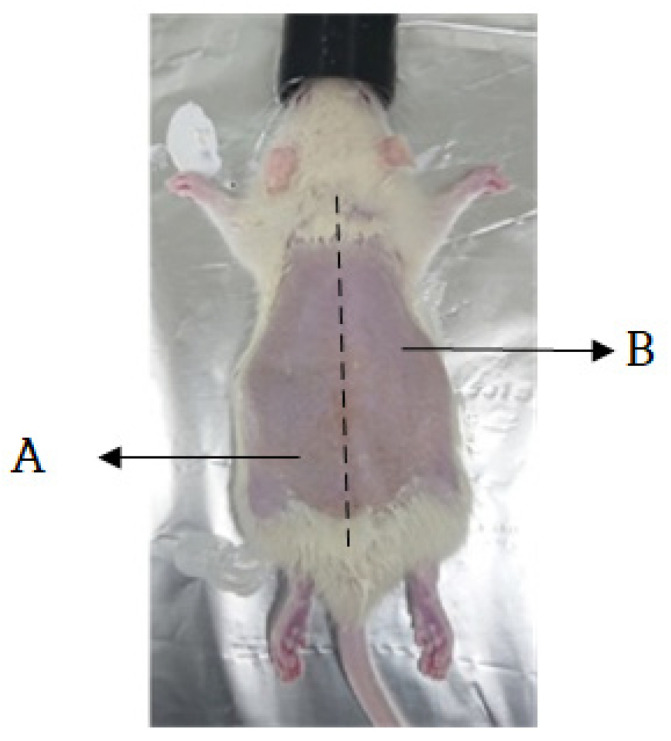
Pressure ulcer site: areas A and B, without and with biophotonic treatment, respectively.

**Figure 5 sensors-23-03620-f005:**
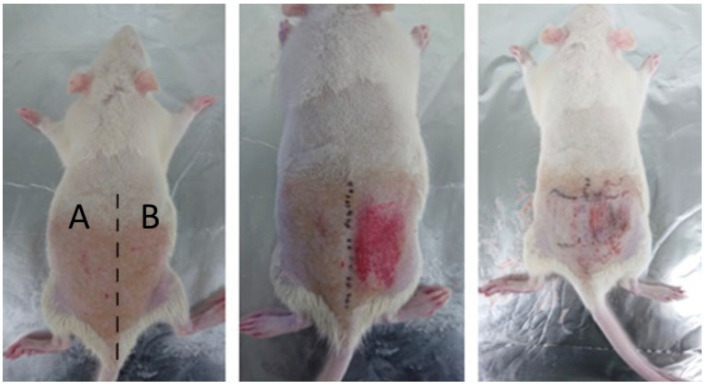
Two-step induction of pressure ulcers in rats through scratches. Stage one and two pressure ulcers were induced in areas A and B, respectively.

**Figure 6 sensors-23-03620-f006:**
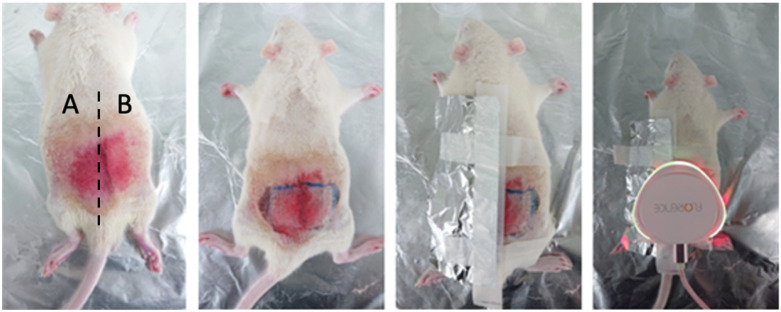
PUCD treatment administration. Aluminum foil was used over area A to restrict the effects of the device over area B. The same criteria for dividing pressure sore-induced areas as shown in Figure 4 were applied.

**Figure 7 sensors-23-03620-f007:**
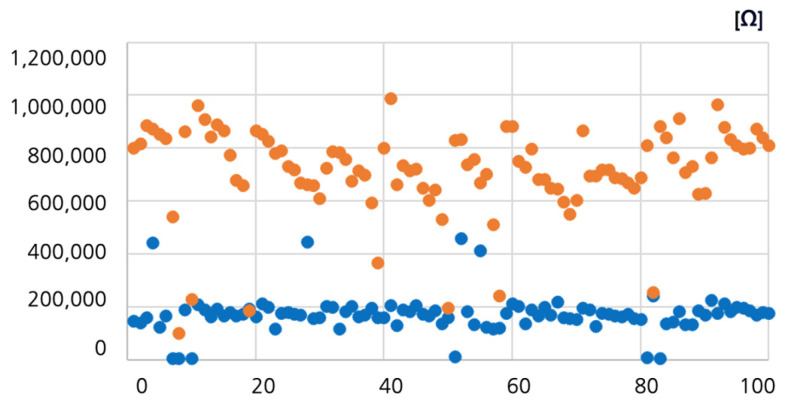
Impedance data after filtering and simple interpolation of measured values using MATLAB for normal and pressure ulcer areas. Blue dots represent pressure ulcers and orange dots represent normal areas.

**Figure 8 sensors-23-03620-f008:**
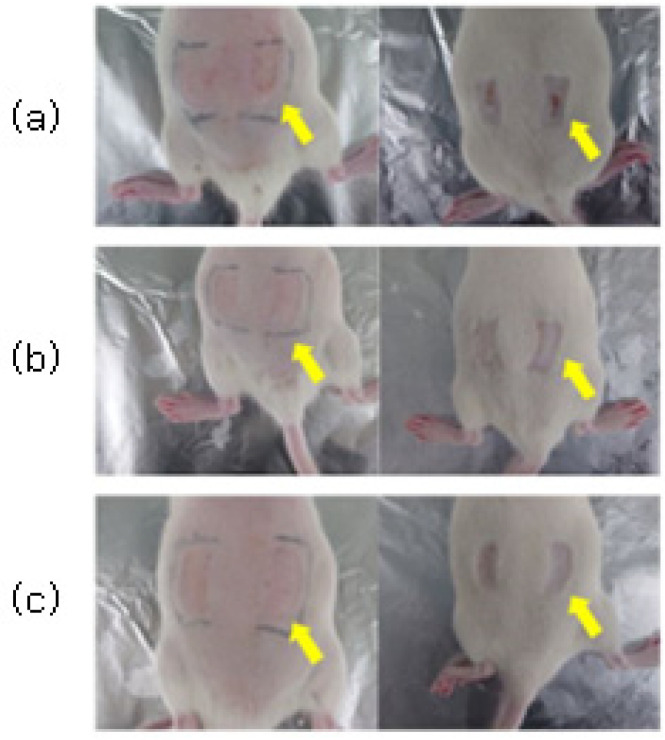
Before and after 7 days with (**a**) no treatment, (**b**) treatment with 50% LED output (360 mW), and (**c**) treatment with 100% LED output (720 mW). You can visually see the actual results in the yellow arrows.

**Figure 9 sensors-23-03620-f009:**
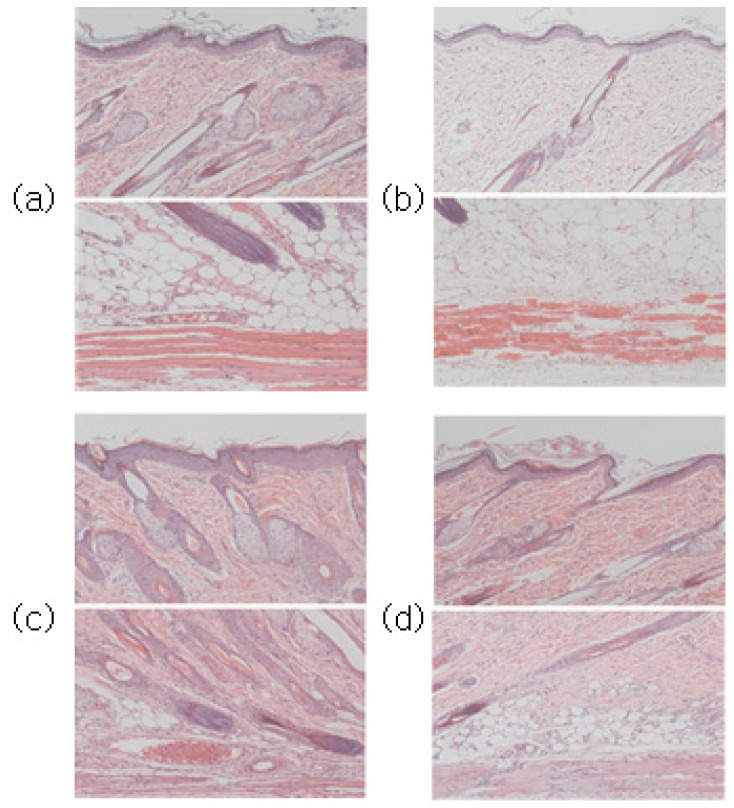
(**a**) Normal group images after H&E and Masson’s trichrome staining at the top and bottom, respectively. (**b**) Stage one pressure ulcer-induced group. Compared to the normal group in (**a**), epithelial loss, dermal tissue arrangement damage, fat layer irregularity, and muscle tissue damage were confirmed. (**c**) Seven days after pressure ulcer induction without any treatment. (**d**) PUCD output was set to 50% (360 mW) for a stage one pressure ulcer.

**Figure 10 sensors-23-03620-f010:**
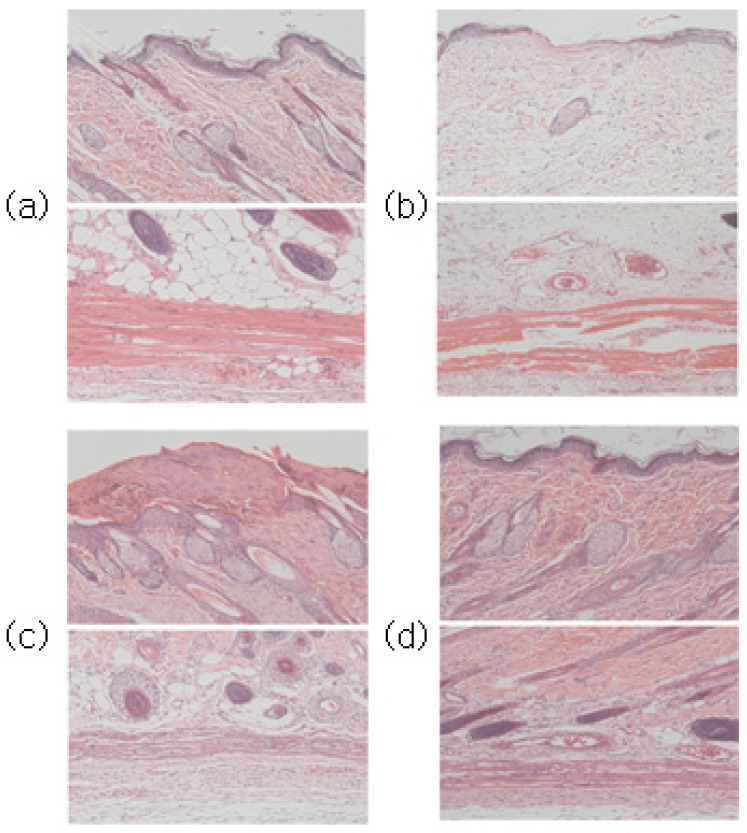
(**a**) Normal group images after H&E and Masson’s trichrome staining at the top and bottom, respectively. (**b**) Stage one pressure ulcer-induced group. Compared to the normal group in (**a**), epithelial loss, dermal tissue arrangement damage, fat layer irregularity, and muscle tissue damage were confirmed. (**c**) Seven days after pressure ulcer induction without any treatment. (**d**) PUCD output was set to 100% (720 mW) for a stage one pressure ulcer.

**Figure 11 sensors-23-03620-f011:**
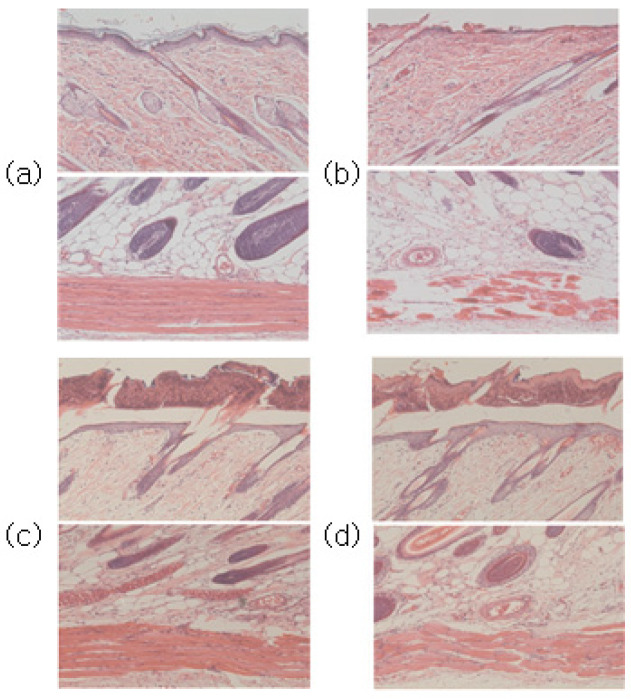
(**a**) Normal group images after H&E and Masson’s trichrome staining at the top and bottom, respectively. (**b**) Stage two pressure ulcer-induced group. Compared to the normal group in (**a**), epithelial loss, dermal tissue arrangement damage, fat layer irregularity, and muscle tissue damage were confirmed. (**c**) Seven days after pressure ulcer induction without any treatment. (**d**) PUCD output set to 100% (720 mW) for a stage two pressure ulcer. When compared with (**c**), epithelial tissue regeneration has progressed.

**Figure 12 sensors-23-03620-f012:**
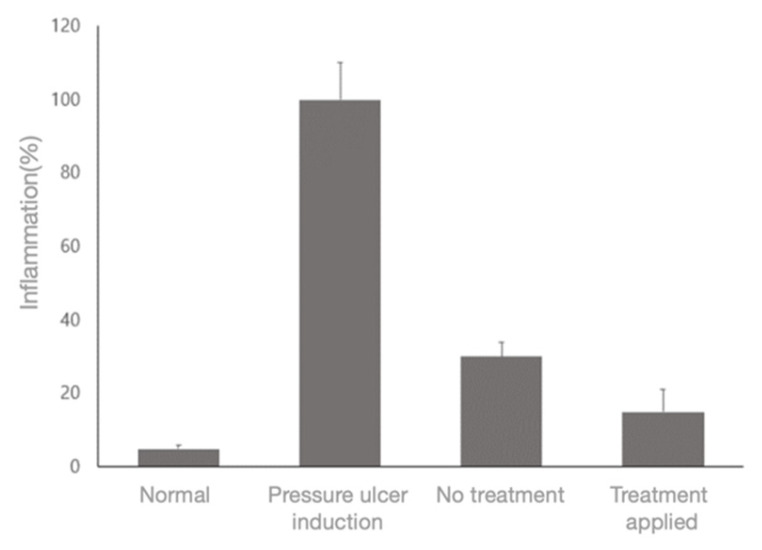
Inflammation-inhibitory effect of the PUCD.

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
