# Peer review of "The Novel Digital Therapeutics Sensor and Algorithm for Pressure Ulcer Care Based on Tissue Impedance"

_sensors, 2023, doi:10.3390/s23073620_

Round 1

Reviewer 1 Report

Dear colleagues!

Thank you for the opportunity to evaluate your research. Taking into account modern ideas about the mechanism of formation of bedsores and measures for their prevention, your work seems interesting and relevant.

Let me ask you some clarifying questions.

1. In the introduction, you almost do not pay attention to the epidemiology of the main diseases responsible for this pathology. What is it connected with?

2. What was the prototype in your study and how was the null hypothesis developed?

3. In figure 1, you need to sign each individual image

4. I can not find the conclusion and permission of the ethical committee to work. Please write about it.

5. In your research results, you write about 3.3. Clinical trials, however, unlike the results of the animal experiment, there are no exact numbers in this part. In the absence of permission to use the technology in humans, this description should be abandoned.

6. In the discussion, I would like to see an analysis of the medical devices market with which you compare your development

Author Response

Thank you for allowing a resubmission of our manuscript, with an opportunity to address the reviewers’ comments.

We are uploading (a) our point-by-point response to the comments (below) (response to reviewers), (b) an updated manuscript with yellow highlighting indicating changes, and (c) a clean updated manuscript without highlights (PDF main document).

Reviewer 2 Report

The manuscript entitle "The Novel Digital Therapeutics Sensor and Algorithm for Pressure Ulcer Cares Based on Tissue Impedance" by Jung et al., describes the development and testing of a pressure ulcer care device (PUCD) that uses a biophotonic sensor to diagnose and treat pressure ulcers. The device was found to have a diagnostic accuracy of 91% and was effective in reducing immune cells in animal models of pressure ulcers. The PUCD was also shown to be effective in identifying immune cells through nuclear shape analysis. The study concludes that the PUCD offers a simple, low-cost, and effective alternative to existing visual diagnosis methods for pressure ulcers, with fewer adverse effects and reduced treatment costs. The authors suggest that mass production of the PUCD could enable improved pressure ulcer care from home.

Before further consideration of the work, the following points should be addressed in detail:

1.      The study only conducted animal and clinical trials, and additional studies with larger sample sizes and longer follow-up periods may be needed to confirm the effectiveness of the PUCD for pressure ulcer diagnosis and treatment.

2.      The animal model used in the study may not fully reflect the human condition, and the results may not be directly applicable to human patients. In particular, the authors claim:  “Going forward, if the PUCD is mass-produced at low cost, improved pressure ulcer care will become possible from home”.

3.      The study does not provide information on the long-term effectiveness of the PUCD treatment, making it unclear how sustainable the treatment is over time.

4.      The study did not compare the effectiveness of the PUCD to other existing pressure ulcer treatment methods, and the cost-effectiveness of the PUCD compared to other treatments is unclear.

5.      The study only reported short-term outcomes, and the long-term effects of the PUCD on pressure ulcer healing and recurrence are unclear.

6.      The study did not report on the potential adverse effects of the PUCD, and further investigation is necessary to ensure the safety of the device.

7.      The study highlights the potential usefulness of impedance spectroscopy for early detection of tissue damage but acknowledges that using it periodically on stage three and stage four pressure ulcers could make wound management difficult due to infection. Further research is needed to address this issue.

8.      The study highlights that pressure ulcer management depends on the expertise of healthcare workers, and while the PUCD may aid newly graduated nurses, it does not replace the need for skilled healthcare professionals.

9.      The study suggests that reducing the number of treatment steps can reduce costs, but it does not provide a detailed cost analysis of the PUCD compared to other treatments.

10.   The article mentions the need for additional research on various optimal treatment conditions, such as using more or less power, but it does not provide a clear roadmap for future research.

Author Response

(The authors gave the same response as above.)

Round 2

Reviewer 2 Report

Thank you for considering comment and suggestions.

I recommend the publication of revised version.